# Involvement of the Intestinal Microbiota in the Appearance of Multiple Sclerosis: *Aloe vera* and *Citrus bergamia* as Potential Candidates for Intestinal Health

**DOI:** 10.3390/nu14132711

**Published:** 2022-06-29

**Authors:** Jessica Maiuolo, Vincenzo Musolino, Micaela Gliozzi, Cristina Carresi, Federica Scarano, Saverio Nucera, Miriam Scicchitano, Francesca Oppedisano, Francesca Bosco, Roberta Macri, Ernesto Palma, Carolina Muscoli, Vincenzo Mollace

**Affiliations:** 1Laboratory of Pharmaceutical Biology, IRC-FSH Center, Department of Health Sciences, University “Magna Græcia” of Catanzaro, Germaneto, 88100 Catanzaro, Italy; v.musolino@unicz.it; 2IRC-FSH Center, Department of Health Sciences, University “Magna Græcia” of Catanzaro, Germaneto, 88100 Catanzaro, Italy; micaela.gliozzi@gmail.com (M.G.); carresi@unicz.it (C.C.); federicascar87@gmail.com (F.S.); saverio.nucera@hotmail.it (S.N.); miriam.scicchitano@hotmail.it (M.S.); oppedisanof@libero.it (F.O.); boscofrancesca.bf@libero.it (F.B.); robertamacri85@gmail.com (R.M.); palma@unicz.it (E.P.); muscoli@unicz.it (C.M.); 3Nutramed S.c.a.r.l, Roccelletta di Borgia, 88021 Catanzaro, Italy; 4IRCCS San Raffaele, Via di Valcannuta 247, 00133 Rome, Italy

**Keywords:** multiple sclerosis, gut microbiota, nutrition, milk, SCFAs, microbiota–brain communication, *Aloe vera*, *Citrus bergamia*, acemannans

## Abstract

Multiple sclerosis (MS) is a neurological and inflammatory autoimmune disease of the Central Nervous System in which selective activation of T and B lymphocytes prompts a reaction against myelin, inducing demyelination and axonal loss. Although MS is recognized to be an autoimmune pathology, the specific causes are many; thus, to date, it has been considered a disorder resulting from environmental factors in genetically susceptible individuals. Among the environmental factors hypothetically involved in MS, nutrition seems to be well related, although the role of nutritional factors is still unclear. The gut of mammals is home to a bacterial community of about 2000 species known as the “microbiota”, whose composition changes throughout the life of each individual. There are five bacterial phylas that make up the microbiota in healthy adults: *Firmicutes* (79.4%), *Bacteroidetes* (16.9%), *Actinobacteria* (2.5%), *Proteobacteria* (1%) and *Verrucomicrobia* (0.1%). The diversity and abundance of microbial populations justifies a condition known as eubiosis. On the contrary, the state of dysbiosis refers to altered diversity and abundance of the microbiota. Many studies carried out in the last few years have demonstrated that there is a relationship between the intestinal microflora and the progression of multiple sclerosis. This correlation was also demonstrated by the discovery that patients with MS, treated with specific prebiotics and probiotics, have greatly increased bacterial diversity in the intestinal microbiota, which might be otherwise reduced or absent. In particular, natural extracts of *Aloe vera* and bergamot fruits, rich in polyphenols and with a high percentage of polysaccharides (mostly found in indigestible and fermentable fibers), appear to be potential candidates to re-equilibrate the gut microbiota in MS patients. The present review article aims to assess the pathophysiological mechanisms that reveal the role of the microbiota in the development of MS. In addition, the potential for supplementing patients undergoing early stages of MS with *Aloe vera* as well as bergamot fibers, on top of conventional drug treatments, is discussed.

## 1. Introduction

Multiple sclerosis (MS) is a neurological and inflammatory autoimmune disease of the central nervous system (CNS), in which selective activation of T and B lymphocytes prompts a reaction against myelin; these cells come from the peripheral circulation, penetrate the CNS and induce an inflammatory cascade resulting in demyelination and axonal loss. MS is characterized by the interruption of immunological self-tolerance to CNS myelin components, resulting in the rupture of myelin sheaths, clear inflammatory infiltrates, the proliferation of astrocytes, gliosis, the activation of microglia, axonal degeneration, oxidative stress, and mitochondrial damage [1]. This neurologic disease predominantly affects young adults, with an incidence of 2.3 million people around the world, and mostly women at a ratio of 2:1 [2,3]. It is responsible for an array of symptoms involving the visual, sensory, motor, and autonomic systems. Optic neuritis (inflammation of the optic nerve) and Lhermitte’s phenomenon are initial and early symptoms characteristic of MS [4,5]. The age of MS onset is around 30 years old, but cases of pediatric or late MS can be found [6,7]. The most common phenotypes of MS are remittent relapsing MS (RRMS), which presents with an acute inflammatory episode and remittance with total or almost total recovery from each seizure. At the base of the inflammation, there is a massive involvement and increase in the lymphocyte population, activation of microglia, oxidative damage, mitochondrial injury, and energy failure [8,9,10,11]. About 15 years after diagnosis, up to 80% of people develop secondary progressive MS (SPMS), with a gradual increase in disability, marked neurodegenerative pathogenesis, and a reduced inflammatory state [12]. Finally, there is primary progressive MS (PPMS), in which progressive disability occurs from the beginning, with massive involvement of the spinal cord, which is manifested by a progressive spastic paraparesis. The trend of disability in the different forms of MS is represented in Figure 1a. The diagnosis of MS requires evidence of plaques in the areas of damage, within the white matter and the concomitant exclusion of other inflammatory, structural, or hereditary conditions that could give a similar clinical picture. In addition, it is characterized by increased latency of visually evoked potentials; the analysis of cerebrospinal fluid, obtained by lumbar puncture, which highlights the presence of oligoclonal bands with B cells that produce immunoglobulins; and magnetic resonance imaging (MRI) showing signal change areas [13]. Until about 10 years ago, MS was considered a two-stage disease that involved early inflammation, responsible for the onset of the disease, in the form of RRMS; and delayed neurodegeneration responsible for the progression of the disease (SPMS), which led to more serious disabilities. Until 2010, the time between the first and the second phase was used for the administration of disease-modifying treatments (DMT) such as interferon beta, glatiramer acetate, and fingolimod (the first oral DMT), with conflicting results that often completely filled the temporal space preceding the progression of the disease and culminating in an irreversible disability [14]. Today, a series of other drugs are available, including three monoclonal antibodies, which are administered earlier in order to avoid wasting that precious time that, from the initial stage, leads to the axonal loss responsible for permanent disability [15]. In Figure 1b, it is reported how the disease could be if it was treated early with the currently validated drugs. The involvement of environmental factors in the onset of MS is increasingly recognized, and this correlation is able to explain the epidemiological increase in the disease. Among these, the gut microbiota seems to have a relevant importance [16]. For this reason, this review has the main purpose of highlighting the existing correlation between the onset of MS and the state of the intestinal microbiota. Specifically, the main knowledge of the types of alteration of the gut microbiota and the evolution of neurodegenerative disease will be specified. In addition, it will be investigated which foods should be avoided in multiple sclerosis, in order not to further alter the intestinal microbiota. Finally, accepting the hypothesis that the onset and/or evolution of autoimmune disease may be related to the health of the gut microbiota, the use of polysaccharides of the *Aloe vera* and *Citrus bergamia* plants could have a marked prebiotic effect.

## 2. Nutrition and MS

Although MS is recognized as an autoimmune pathology of the central nervous system, the specific causes are many and thus, to date, it has been considered a disorder resulting from environmental factors in genetically susceptible individuals [17]. It is precisely this variability of the causes of MS that determines the partial effectiveness of the drugs in use, each with a distinct immunomodulatory mechanism. Among the environmental factors hypothetically involved in MS are some viral infections, ethnicity, exposure to tobacco smoke, organic solvents, toxins or heavy metals, sun exposure, poor levels of vitamin D, obesity in adolescence, latitude and diet [18]. Nutrition seems to be related to MS, although the role of nutritional factors is still unclear and further clinical trials are needed [19]. The scientific literature obtained prior to 2005 demonstrated that the only correlations between nutrition and MS concerned the intake of macronutrients [4]. For example, for a person affected with MS, an adequate intake of carbohydrates is very important because it allows them to maintain good energy levels and also counteract fatigue, a general symptom of the disease. Carbohydrates provide the energy needed to keep the body active without affecting sugar reserves (in the form of glycogen stored in the liver) and fat stocks. However, it is essential to reduce the intake of refined simple sugars, which increases the caloric content of food. Precisely for this reason, the diet of the patient with MS should never miss out on complex carbohydrates that provide energy by replacing them with protein to build muscle mass [20]. Protein intake is important for people with MS for three reasons: (a) remedy for significant weight loss; (b) compensation for the decrease in muscle mass; (c) prevention of the possible formation of ulcers from decubitus, in the case of physical immobility [21]. An excessive intake of saturated fats of animal origin could worsen the course of MS, alter the stability of the myelin sheath favoring demyelination [22,23]. Saturated fatty acids (SFAs) of animal origin must be controlled to avoid the increase in inflammatory processes in MS, and a low-fat diet may have some beneficial effects in the disease [24,25,26]. Intake of SFAs increases LDL cholesterol, and this phenomenon is associated with a worsening of MS due to a direct impact on the immune system, activation of proinflammatory toll-like receptors, and increase in the proinflammatory transcription factor NF-kB [27]. In a study using an animal model of MS, mice were fed a high-fat diet and increased T cells, inflammatory cytokine expression (IL-1β, IL-6, and IFNγ), and infiltrating macrophages were detected [28]. The length of the fatty acid chain seems to be an extremely important factor in determining the progress of MS. In fact, while long-chain fatty acids promote the differentiation of proinflammatory T cells (TH1 and TH17) through the members of the MAP kinase family (an important action is carried out by P38 and JNK1), short-chain fatty acids (SCFAs) promote the differentiation of regulatory T cells through the production of anti-inflammatory cytokines [29]. Recently, a study was published in which the correlation between fat intake and MS was highlighted: 219 young patients with a high intake of saturated fats had a threefold higher risk of developing new lesions compared to MS patients who did not consume as much fat [30]. To date, it is known that minerals, antioxidants, trace elements, and vitamins can also be related to MS [31]. For example, dietary antioxidants have important biological consequences in MS, since oxidative stress is one of the most important components of the inflammatory process, leading to the degradation of myelin and axonal damage [32]. A recent study demonstrated that the consumption of foods with anti-inflammatory properties reduced the biological synthesis of proinflammatory molecules and improved the effectiveness of drugs with immunomodulatory activity [33]. Polyunsaturated fatty acids (PUFAs) are characterized by multiple double bonds within the fatty acid chain and are found in foods such as fish, flax seed, and walnuts. PUFAs reduce inflammation through conversion into the anti-inflammatory prostaglandins E1 and E2, with effects on cytokine production, leukocyte migration, and other immune system components [34]. In vivo studies have demonstrated that the administration of PUFAs was able to reduce the production of inflammatory cytokines, prevent demyelination, and promote remyelination [35,36]. Scientific studies have demonstrated, in vivo, that even a diet rich in salt determines adverse effects in MS: for example, a high amount of salt promotes the differentiation of proinflammatory TH17 cells, which develop a more pathogenic phenotype and a worsening of the course of the disease. In addition, Farez et al. found, in a study of 70 patients with RRMS, that those with a medium–high salt intake had clinical recurrence rates 3.95 times higher than those with low sodium intake [37]. However, it is worth pointing out that these findings have been contradictory and that, currently, there are no further published clinical studies on the correlation of sodium intake and MS. On the contrary, increased fruit and vegetable intake has been associated with reduced levels of disease activity and disability [38]. More specific studies have demonstrated that a diet rich in animal fat, milk, dairy products, meat, hydrogenated fats, and sugars and low in fruit, vegetables, and whole grains was related to a higher prevalence of MS and a higher level of disability [39,40]. In particular, an interesting study was conducted on 20 MS patients divided into two groups: 10 patients who consumed a diet characterized by high vegetable content and reduced protein (HV/LP), compared to 10 patients who ate a typical Western diet (WD) for a period of 12 months. The results obtained demonstrated that the HV/LP group had a reduction in proinflammatory T cells, a reduction in proinflammatory interleukins, and an increase in anti-inflammatory T lymphocytes [41]. The pathophysiology of MS indicates that there are three factors on which to act in order to influence the course of the disease: (1) modulate inflammation; (2) protect against neurodegeneration; and (3) promote repair of the nervous system. It has been demonstrated that diet exerts a systemic influence on all three of these pathways, resulting in more or less beneficial effects [42,43]. A Mediterranean-style diet with a low content of saturated fat and processed foods, many polyunsaturated and monounsaturated fats, especially fish and olive oil, and plentiful fruits and vegetables has been associated in MS with reduced disability [44], neurodegeneration [45], and cognitive decay [46]. In recent years, a close correlation between MS and vitamin levels has been highlighted. For example, the association between MS and vitamin D deficiency suggests that this vitamin may play a role in the immune response [47]. Vitamin D, whose known forms include D2 or ergocalciferol and D3 or cholecalciferol, is taken either through exposure of the skin to sunlight or through the diet. The largest study related to this topic was carried out on over 7 million American soldiers and demonstrated an inverse correlation between serum levels of vitamin D and the risk of developing MS [48]. Moreover, vitamin D seems to be not only a risk factor for the onset of MS, but also able to modulate the activity of the disease and its progression. In fact, low serum levels of vitamin D have been associated with an increase in disability, increased rate of recurrence, and an increase in the load of lesions, as evidenced by MRI [49]. The pathophysiological mechanism of vitamin D responsible for the onset or progression of MS seems to be its role in the activation and proliferation of lymphocytes, the differentiation of T-helper cells, and its regulatory effects on the immune response [50]. Vitamin D supplementation led to a reduction in CD4+ T cells producing IL-17, and the inhibition of the proliferation of B cells by induction of the apoptotic process [51]. Several studies demonstrated the influence of other vitamins on MS: plasma concentrations of vitamin B12 and folate were decreased in patients with MS, due to their role in the formation of the myelin sheath [52]. Finally, Bitarafan et al. determined that treatment with vitamin A improved cognitive ability and reduced disability in MS [53]. The impact of various dietary factors in MS is very interesting and, for this reason, further preclinical models, epidemiological research, and prospective and clinical studies would be desirable. Therapy for MS cannot be replaced by a particular diet, but a healthy nutritional intervention can improve patients’ physical and inflammatory state. Figure 2 reports the effects of foods on inflammatory processes in MS.

### Milk, Dairy Products, and MS

The exact mechanism for instigating an autoimmune response is unclear, but the concept of molecular mimicry, understood as a significant similarity shared by foreign antigens and self-proteins, offers a viable hypothesis to explain the activation of specific autoreactive T cells [54,55]. It has already been demonstrated that the molecular mimicry hypothesis is well linked with MS. In fact, antigen sequences of some viruses have been demonstrated to stimulate myelin basic protein (MBP)-specific CD4+ T cell clones: there is, presumably, a structural homology between viral epitopes and MBP [56,57]. To confirm this concept, important experiments have been conducted in which the sequence similarity between some viruses and myelin proteins has been demonstrated in autoimmune encephalomyelitis (EAE), an autoimmune disorder widely used as an experimental animal model of MS [58]. Among common foods, milk and dairy products have been indicated as an etiological factor in MS since 1970 [59,60]. Milk is considered a complete food necessary and sufficient for the growth and development of newborn individuals. Its composition changes in relation to the species to which it belongs. The composition varies according to the stage of lactation and between full-term and preterm infants [61]. The average composition of macronutrients in mature human milk is estimated to be between 0.9 and 1.2 g/dL for proteins, between 3.2 and 3.6 g/dL for fats, and between 6.7 and 7.8 g/dL for lactose. As for the energy value, it is from 65 to 70 kcal/dL, but is highly correlated with the fat content [62]. The mechanism by which milk could trigger autoimmunity through a response to CNS antigens, contributing to the development of MS, is not fully clarified, but high cell reactivity has been reported against several cow’s milk proteins in MS patients [63]. Another milk protein involved in molecular mimicry is butyrophilin (BTN) [64]. Stefferl et al. demonstrated a sequence homology between the antigen of a myelin oligodendrocytic glycoprotein (MOG) and butyrophilin [65]. It was later demonstrated that antibodies specific to the extracellular domain of MOG cross-reacted with bovine milk protein BTN in a mouse model of MS [66]. BTNs are transmembrane glycoproteins present in milk fat cells and constitute a large family of structurally similar transmembrane type 1 proteins belonging to the immunoglobulin superfamily. In humans, the *BTN* gene family encodes 14 proteins, and members of the BTN family are characterized by a similar domain organization, with classical butyrophilin consisting of three domains: two extracellular Ig-like domains, IgV and IgC, one transmembrane domain (TMD), and the cytoplasmic B30.2/SPRY domain [67]. It has been demonstrated that the subdomain of *N*-terminal bovine BTN has a striking similarity (50% sequence homology) with the glycoprotein of oligodendrocytes of human myelin MOG [68]: therefore, it is reasonable to assume that molecular mimicry occurs and that the intake of BTN, by milk consumption, can determine an immune response against MOG [69]. In this way, molecular mimicry is able to influence the course of MS, following interaction with BTN present in milk and dairy products. Figure 3 shows the common structure between BTN and MOG.

## 3. Gut Microbiota and Brain

The gut of mammals is home to a microbial community of about 2000 bacterial species called the microbiota, whose composition changes over the life of each individual. To date, it is known that microorganisms that reside in the intestine exceed human somatic cells by a ratio of 10:1 and the microbial genome is composed of about 3 × 10^6^ genes, 150 times the length of the human genome [70]. Factors responsible for changing the composition of the microbiota include incorrect nutrition, pH, oxygen level/redox state, availability of nutrients, water activity, temperature, drug therapy, pathological conditions, sleep disturbance, and drug abuse, among others [71]. In healthy adults, the microbiota is primarily composed of five bacterial phyla, *Firmicutes* (79.4%), *Bacteroidetes* (16.9%), *Actinobacteria* (2.5%), *Proteobacteria* (1%), and *Verrucomicrobia* (0.1%) [72], and when the gut microbiota has a high diversity and abundance of microbial populations, this condition is known as eubiosis. On the contrary, the state of dysbiosis refers to altered diversity and abundance of the microbiota [73]. A co-metabolism is generated between the microbiota and the host: this relationship is symbiotic and mutually beneficial; the host provides a suitable habitat for the microbiota and nutrients, while the intestinal microbiota supports the development and intestinal maturation of the host [74]. In the case of eubiosis, the human gut microbiota plays certain roles in maintaining health, including the breakdown of food substances to release nutrients that would otherwise be inaccessible to the host, protection against the colonization of pathogens, and stimulation and/or modulation of the immune system [75]. Today, it is known that the microbiota is able to control and influence many segments of the host such as the immune system, the digestive system, and the brain [76,77,78]; a real crosstalk exists between the microbiota and the immune system of the host, helping to promote tolerance to the harmless antigens of the microbiota [79]. The microorganisms present in the first phase of intestinal colonization are predominantly aerobic, including *Staphylococci*, *Enterobacteria*, and *Streptococci*; subsequently, they become predominantly anaerobic and without pathogenic potential. The dietary factor is fundamental for the composition of the microbiota: in fact, during the first year of life, the intestinal microbial colonization depends strictly on the type of milk the child is fed, the type of weaning it undergoes, and the different types of food it consumes. However, it is also important to assess the use of antibiotic therapies that, if prolonged, completely revolutionize the enrichment of the intestinal microbial composition [80]. There is a substantial difference in the microbial composition of breastfed babies (BF) and those taking various milk formulations (FF): BF infants have a more uniform intestinal microbial population than FF infants, suggesting that breast milk can positively influence the composition of the microbiota [81]. It has been shown that BF infants have a more uniform and equilibrated microbial intestinal population than FF infants, and this condition guarantees a better future state of health: in fact, BF children have proper development of the immune system, improved immune tolerance, and reduced incidence of allergic, inflammatory, and autoimmune diseases [82]. At the same time, FF infants demonstrate an increased incidence of developing inflammatory bowel disease and impaired neurodevelopment [83]. The increased protection conferred by breast milk could be explained by its recognized composition in proteins, fats, carbohydrates, immunoglobulins, endocannabinoids, and indigestible polysaccharides. Some polysaccharides act as prebiotics capable of selectively stimulating the growth of beneficial bacteria; moreover, maternal immunoglobulins give the correct protection and direct optimal development of the microbiota [84]. Even the formulated milk is composed of the same macronutrients, but obviously in different ratios, and this could explain the increased intolerance towards its proteins. The composition of the microbiota still varies after three years of life and, in the absence of prolonged use of antibiotics and drugs or radical changes in dietary regimens, it remains fairly stable until old age: *Bifidobacteria* decrease while *Bacteroidetes* and *Firmicutes* increase [85]. Dysbiosis of the intestinal microbiota is closely related to various diseases such as type 2 diabetes mellitus, hypertension, obesity, necrotizing enterocolitis, and inflammatory bowel disease [86], and now it is evident that there is also a link with the central nervous system [87,88]; for this reason, the intestine is sometimes called the “second brain” [89]. In particular, microbiota dysfunction can play a key role in the development of certain neurological diseases and appropriate intervention to correct the integrity of the microbiota, which can have a positive influence on the course, symptoms, and clinical conditions of many neurological diseases [90,91].

### 3.1. Microbiota–Brain Communication

Many recent scientific papers have suggested that there is a close correlation between the gut microbiota and the brain, and the gut microbiota is responsible for bidirectional interaction with the central nervous system [92,93]. The gut microbiota, for example, is able to synthesize many neurotransmitters, including serotonin, dopamine, norepinephrine, and δ-amino butyric acids (GABA), that also have effects on the brain. In fact, *Bifidobacterium infantis* has been demonstrated to elevate plasma tryptophan levels and thus influence central serotonin transmission; *Escherichia bacillus*, and *Saccharomyces* spp. can produce noradrenaline; *Candida*, *Streptococcus*, *Escherichia*, and *Enterococcus* spp. can produce serotonin; and *Lactobacillus* and *Bifidobacterium* can produce GABA [94]. Since these neurotransmitters, despite entering the bloodstream, are not able to cross the blood–brain barrier, they could act directly on the neurons of the enteric nervous system and communicate with the brain using the same “language” as in the central nervous system [95]. In addition, the gut microbiota can produce fundamental metabolites such as short-chain fatty acids (SCFAs), which include butyrate, propionate, and acetate, and are able to influence the energy balance and metabolism of the brain, as well as possessing neuroactive properties [96]. Finally, the gut–brain axis also involves immunity through cytokines, which can be produced in the intestine and reach many areas of the brain [97].

### 3.2. Gut Microbiota and MS

Many studies report that the intestinal microbiota is able to model immune responses outside the intestine. For example, in all the models used that prove the involvement of the microbiota in autoimmune arthritis [98] and experimental autoimmune encephalomyelitis [99], there is always an excessive cellular response of T helper 17 cells [100]. Mazmanian and colleagues demonstrated, in an EAE experimental model, that a germ-free state reduced inflammation and clinical pathology compared to colonized mice, demonstrating that the microbiota is able to modulate the onset and severity of autoimmune encephalitis [101]. Many studies on animal models have demonstrated that there is a relationship between the type of intestinal microflora and the progression of MS. In particular, patients with MS have a reduction in the proportion of *Faecalibacterium* and *Fusobacterium* and an increase in *Shigella*, *Clostridium*, *Eubacterium rectal*, *Escherichia*, *Firmicutes*, and *Corynebacterium* [102,103]. The main mechanisms involved in the occurrence of MS concern some metabolic by-products of the intestinal microflora that are responsible for the coding of the protein FOXP3, a transcriptional regulator, that binds to the promoters of the genes involved in the development and regulation of T-cell receptors and promotes the attenuation of the immune response [104]. These by-products of the microbiota include SCFAs, which are directly responsible for activating the FOXP3 pathway and modulating the immune response. In conditions of normality and health, the regulatory mechanism works perfectly, but in the presence of intestinal dysbiosis, the regulatory processes are altered and the pathways that lead to autoimmunity are activated [105]. *Bacteroidetes* of the gut microbiota are able to produce lipid 654, which, after binding to the human and mouse Toll-like receptor 2 (TLR2), preserves the functioning of the immune system. It was demonstrated that lipid 654 was present in the serum of all healthy subjects examined. On the contrary, patients with MS had extremely low lipid 654 levels, indicating it, for the first time, as a serum biomarker of MS [106,107]. Tsunoda et al. have demonstrated that some microbiota bacteria, such as *Clostridium perfringens*, produce natural toxins that are involved in the early stages of MS [108]. In particular, these toxins are absorbed by the intestine, enter the bloodstream, and cause the typical symptoms of the initial stages of MS, such as blurred vision, lack of coordination, or spastic paralysis [109]. Already in the 1990s, there was a suspicion that these toxins could be a potential cause of MS: people, in fact, are not a natural host of *Clostridium perfringens*, but became so in the case of intestinal dysbiosis, making this bacterial family prevalent [110]. It has been hypothesized that toxins produced in the microbiota can be transmitted into the brain and affect myelin and nonmyelin areas [111]. Finally, it is important to note that even some first- and second-line drugs prescribed for MS alter the composition of the intestinal microbiota. For example, the drug glatiramer acetate induces a reduction in *Lactobacillaceae*, *Bacteroidaceae*, *Faecalibacterium*, and *Clostridium* compared to untreated patients [112]. Figure 4 highlights the direct correlation between the gut and brain and shows how the microbiota exerts its action on the brain.

## 4. The Interaction of Plant Fibers with the Intestinal Microbiota

The ability of the diet to modify the gastrointestinal microbiota of humans and other mammals has been extensively investigated. In addition to a description of the effects of the main macronutrients on the gut microbiota, a further study should concern the intake of fiber and how quickly the composition of the microbiota changes. For example, a major change in bacterial diversity has already been demonstrated in humans 24 h after the intake of a fiber-rich (>30 g/day) agrarian diet instead of a fiber-free, meat-based diet [113]. The Codex Alimentarius Commission defines dietary fiber as “carbohydrate polymers with 10 or more monomeric units, which are neither digested nor absorbed in the human small intestine.” The role of dietary fiber in the gut microbiota depends on several factors including the origin, chemical makeup, physical structure, and degree of polymerization (chain length) [114]. The physicochemical properties of fiber include fermentability, solubility, and viscosity. Insoluble fiber, such as cellulose, is generally poorly fermented by intestinal microbes. This fiber increases the rate of intestinal transit and thus reduces the amount of time available for the bacterial fermentation of undigested food [115]. However, their high solubility and viscosity results in unique therapeutic effects, including the ability to improve glycemic control and lower blood cholesterol levels [116]. Among the highly soluble, fermentable, and viscous fibers are β-glucan and pectins [117]. Soluble and nonviscous fibers are readily fermented from the gastrointestinal microbiota and include inulin, resistant maltodextrins, resistant starch, polydextrose, and soluble corn fiber [118]. The different solubility of complex carbohydrates affects the location of fermentation in the human gastrointestinal tract: for example, soluble fiber and pectin are metabolized by bacteria in the ileum and ascending colon; fiber that is less soluble, such as cellulose, can be partially fermented in the distal colon [119]. In general, fermentable fiber behaves as a prebiotic [120], although it is appropriate to add that not all fiber can be classified as a prebiotic. To date, we know that the consumption of prebiotics is a dietary strategy that allows the intestinal microbiota to be modified to improve health. The definition of a prebiotic has evolved over the years; the currently accepted and most comprehensive one is “a non-digestible food ingredient which selectively stimulates its fermentation by the microbiota, increases the growth and/or activity of a limited number of bacteria in the gastrointestinal tract, and positively affects the host. It must also be resistant to gastric acidity and hydrolysis by enzymes during gastrointestinal absorption” [121]. Numerous scientific evidences have highlighted the role of glyconutrients such as dietary fibers, useful to improve the composition of the intestinal microbiota [122]. In particular, Ambratose is a combination of eight sugars (acemannan), that the body uses to produce glycoproteins, and has been used in humans for many years, given its antioxidant effects [123], the well-known immune boosting benefits and the improvement of cognitive performance [124]. It has been demonstrated in vitro that Ambratose is a very promising prebiotic and clinical data, conducted on 350 participants, demonstrated that treatment with Ambratose (2–4 g per day for 6 months) did not produce adverse events, improving overall intestinal health [125,126]. Given their high proportion of acemannans and fibers, we propose *Aloe vera* and *Citrus bergamia* as hypothetical prebiotic intestinal remedies.

### 4.1. Involvement of Aloe vera with the Intestinal Microbiota

*Aloe vera*, scientifically known as *Aloe barbadensis*, belongs to the Lilaceae family and includes over 360 species. This plant is a perennial succulent xerophyte, native to Africa but present in various regions of the world characterized by a subtropical climate [127]. Its use has been known in folk medicine for over 2000 years, and even today it remains an important component in the traditional medicine of China, Japan, India, and the West Indies because of its countless beneficial effects on human health, including immunomodulatory, anti-diabetic, hepatoprotective antioxidants, anticancer, antiviral, and antibacterial effects, among others [128]. In general, succulents have adapted to live in areas with low water availability and are characterized by tissues with high accumulation of water. For example, *Aloe vera* is 99–99.5% water, while the remaining portion (0.5–1%) is composed of solid materials including minerals, enzymes, water-soluble and fat-soluble vitamins, simple/complex phenolic polysaccharide compounds, and organic acids [129]. The most used organ of *Aloe vera* is the leaf, which appears triangular and is 20 × 5 inches (length and width). The leaf appears to consist of three layers: (1) the outer part is the rind, about 15–20 cells, has a protective function, and synthesizes carbohydrates and proteins; (2) the intermediate section is a bitter and yellow latex composed of glycosides and anthraquinones. The molecules of this layer are responsible for the laxative effects of *Aloe vera*. (3) The innermost portion is visible as a clear gel, also known as the mucilaginous layer, and is composed of 95% water, lipids, amino acids, sterols, vitamins, and glucomannans. In particular, dried aloe gel contains about 55% polysaccharides, 17% sugars, 16% minerals, 7% proteins, 4% lipids, and 1% phenolic compounds [130]. The vascular bundles contained in the peel consist of xylem and phloem. Xylem participates in the transport of raw lymph (water and mineral salts) from the roots to the leaves. Conversely, the phloem deals with the transport of the elaborated lymph (water, starch, and other small organic molecules) to all parts of the plant and the roots [131]. Figure 5 shows the three main sections of the *Aloe vera* leaf.

*Aloe vera* gel is composed of 75 active ingredients, which are recognized as the main healing elements of the plant. Although the main properties are attributed to the polysaccharides present in the inner leaf, today it is believed that the effects should be attributed to the synergistic action of all compounds. *Aloe vera* gel consists, as already mentioned, mainly of water, but we also find a high percentage of polysaccharides; among these, acemannans are the most important [132]. Acemannans are glucose-bound mannose units linked by β-(1→4) bonds constituting the backbone of the polysaccharides. Repetitive units of glucose–mannose exist at a ratio of 1:3, but other ratios have been reported (1:6, 1:15, and 1:22). These discrepancies in glucose–mannose ratios are visible in different species, depending on the seasonality of the plant from which they are obtained and the different methods of the treatment of samples [133]. The molecular weights of these polysaccharides vary between 30 and 40 kDa. However, in some circumstances, they can reach 1000 kDa [134]. Carbohydrates, such as mannans, galactans, arabinans, arabinogalactans, and pectic substances, are indigestible, and fermentable fiber could have prebiotic effects. In general, prebiotics, initially defined as nonviable food components that confer a health benefit to the host associated with modulation of the microbiota, are carbohydrate-like compounds that can be used to support or modify the composition of the microbiota for the benefit of human health [135]. Probiotics, on the other hand, are a group of microorganisms that work in close contact with the intestinal microbiome [136]. Recently, scientific studies have suggested that, by taking both prebiotics and probiotics, one can fight chronic diseases, since colon fermentation produces SCFA [137]. Liu et al. have recently demonstrated that, following the administration of *Aloe vera*, the content of SCFAs, derived from the fecal fermentation of plant polysaccharides, has increased significantly. In addition, microbiota abundance has also increased, as demonstrated by the presence of *Prevotella*, *Catenibacterium*, *Lachnospiraceae*, and *Coprococcus* and the simultaneous reduction of harmful microbiota including *Escherichia*–*Shigella* and *Veillonella*. Finally, the polysaccharides of *Aloe vera* have been demonstrated to enhance the metabolism of fructose and increase the gene expression of enzymes capable of depolymerizing polysaccharides [138,139]. Another recent study evaluated in vivo the effects of polysaccharides of *Aloe vera* on general health, serum biochemistry, and intestinal SCFAs production in mice. The results demonstrated that mice with supplementation for 14 days had good health comparable to those of the control group, and that the production of cecal and colon SCFAs significantly increased [140]. These results deserve further study and support the possible incorporation of *Aloe vera* mucilage in the development of a variety of prebiotic food products in order to improve gastrointestinal health and reduce the incidence of MS, whose development is related to intestinal dysbiosis.

### 4.2. Involvement of Citrus bergamia in the Intestinal Microbiota

The Mediterranean area is rich in fruit, herbs, spices, and vegetables used in traditional medicine for the treatment of several diseases. The Rutaceae are a family of dicotyledonous angiosperm plants that includes about 1600 species, mostly woody (but also some herbaceous), characterized by the presence of oleiferous glands that produce aromatic ethereal oils. Belonging to this family are citrus fruits (genus *Citrus*), and the regions suitable for the cultivation of these types of fruits are tropical and subtropical areas because of their clayey–calcareous soil and favorable climatic conditions (predominantly tropical and subtropical) [141]. The most important citrus fruits are sweet orange (*Citrus sinensis*), tangerine or mandarin (*Citrus reticulata*), lemon (*Citrus limonum*), lime (*Citrus aurantifolia*), bergamot (*Citrus bergamia*), and grapefruit (*Citrus vitis*) [142]. In particular, bergamot is an endemic plant (Calabria, Italy) that produces a yellow fruit, which could be considered a subspecies of bitter orange or a hybrid derived from bitter orange and lemon [143]. From the peel of bergamot, it is possible to obtain an essential oil, which has significant properties [144] in many areas, including melanogenic [145], anticancer [146], painkiller [147], neuroprotective [148], antioxidant [149], and antibacterial [150]. The pulp and juice of bergamot are a rich source of polyphenols, in particular flavonoids, and are used in the form of nutraceuticals with countless beneficial properties such as anticancer [151], antiviral and anti-inflammatory [152,153], antioxidant, cardioprotective [154], lipid-lowering [155,156], metabolic-protective [157], and antiplatelet [158]. Bergamot, like most citrus fruits, is rich in vitamins, minerals, and dietary fiber, which are important for human nutrition, growth, and development [159]. Today, it is estimated that the processing of citrus is mainly focused on the production of fruit juices and essential oils and that only 33% of citrus fruits are used. The remaining part, about 70%, is converted into waste [160]. Nevertheless, large quantities of citrus waste are rich sources of secondary metabolites, flavonoids, polyphenols, carotenoids, essential oils, dietary fiber, sugars, ascorbic acid, and some trace elements [161]. Bergamot contains many flavonoids, including naringin, neohesperidin, neoeriocithrin, neodiosmin, and eriodictyol, whose concentrations are higher than in other citrus fruits [162,163]. In addition, glycosidic onjugates of neohesperidin and naringin, brutieridin and melitidin are found in higher quantities in bergamot but not in oranges and lemons [164]. This composition makes the bergamot flavonoid profile unique, justifying its beneficial properties for human health. Interestingly, these flavanones have been demonstrated to influence intestinal bacterial growth, gene expression, and the increase in beneficial bacteria for the production of SCFAs, behaving as prebiotics [165,166]. Since the dietary intake of fiber confers a beneficial effect on the gut microbiota, as explained above, and bergamot fiber has been demonstrated to be responsible for improved lipid metabolism [167], we wonder what the effect of the albedo fibers of bergamot—currently considered waste material—would be on the gut microbiota and whether they can be considered prebiotics. The albedo of bergamot fruit represents a source of selected fiber that could contribute to increasing the bulk in stool—an effect that helps to cause movement of the intestines. Moreover, bergamot fibers are rich in antioxidants and nutrients able to improve digestive disorders. If this hypothesis were proven, the beneficial role of bergamot would be supported, and a healthy profit could be derived from its waste products. To this end, therefore, it will be important and necessary to carry out appropriate experiments. In Figure 6 a representation of bergamot fruit and albedo fibers are shown.

## 5. Discussion

MS is a neurodegenerative and autoimmune disorder of the nervous system in which massive activation of T and B lymphocytes occurs. This condition causes a constant inflammatory state and a specific reaction against “self” myelin, resulting in demyelination, axonal loss, and increasing disability. Although this is the correct definition for MS, today it is believed that it can also be defined as a pathology resulting from environmental factors in genetically predisposed individuals [168]. In fact, since genetic predisposition only partially explains the increased risk of MS, lifestyle and environmental factors strongly influence the risk of developing the disease. In particular, some viral infections, exposure to tobacco smoke, alcohol, ethnicity, exposure to organic solvents, toxins, heavy metals, poor levels of vitamin D, sun exposure, obesity in adolescence, and diet may affect adaptive and/or innate immunity, and promote the autoimmunity of the disease [169]. It has become increasingly accepted that nutrition is closely related to MS, although it has not been clarified how important food intake is for the aggravation of the pathology. What is known is that patients with MS, in addition to drug treatment, should follow a particular dietary scheme [170,171]. As for all inflammatory pathologies, it would be advisable to take in moderation proinflammatory foods such as refined sugars, saturated fats, and salt [172,173]. Exposure to toxins and heavy metals should be minimized [174], and patients should take nutraceuticals with recognized anti-inflammatory and neuroprotective function [175,176] and avoid the alteration of the blood–brain barrier [177,178]. The consumption of milk and dairy products is particularly harmful for patients with MS [179]. Humans are the only mammals that drink milk after childhood—that is, after a great reduction or suspension of the expression of the lactase enzyme, allowing the cleavage and digestibility of lactose [180]. The reasons for a pathological reaction to milk in patients with MS could be many: on the one hand, the lack of lactase could explain the lactose intolerance followed by cross-reactions to the components of milk [181,182]. On the other hand, adverse reactions to milk proteins, casein, BSA, etc., may occur. Over the years, another interesting hypothesis has also been confirmed by experimental data, which have demonstrated that there is a phenomenon of molecular mimicry between the glycoprotein MOG of oligodendrocytic myelin and butyrophilin (BTN), a transmembrane glycoprotein present in milk fat cells. In particular, MOG and BTN present a marked sequence homology that is able to explain the molecular mimicry and the subsequent immune response against MOG following the intake of milk and dairy products [77]. Finally, there is another hypothesis that might explain the milk–MS connection (or, more generally, the food–MS connection) and it is to be sought in the presence of the gut microbiota. Our bodies are colonized by trillions of microbial cells whose coordinated actions are considered important for human life. Most live in the gut, constituting the so-called gut microbiota, consisting of a number of cells about 10 times that of somatic eukaryotic cells and constituting a weight of more than 1 kg [183]. To date, it has been recognized that there is a marked correlation between the alteration of the intestinal microbiota that is formed during childhood, and the onset of immune and metabolic pathologies in adult life [184,185]. As a result, the need to maintain good microbiota quality in childhood is clear, so as to reduce the likelihood of developing certain diseases later on [186]. Achieving a balanced gut microbiota could be facilitated by the development of strategies that improve the formation, composition, and activities of the microbiota—for example, the use of nutraceutical products (such as probiotics and/or prebiotics) is spreading [187]. Initially, it was a common opinion that the colonization of the infant intestine began at birth. Today, it is known that there is also a bacterial presence in the placenta, umbilical cord, and amniotic fluid. Even the intestinal colonization of the fetus could be affected by the maternal microbiota [188]. The development and maturation of the gut microbiota constitute a dynamic and nonrandom process, and depend on the type of childbirth, type of feeding, use of antibiotics, nutrition, age and metabolic status of the mother, family genetic predisposition, and lifestyle [189]. For example, the mode of delivery to which the pregnant woman is subjected provides important elements of variability in the intestinal colonization of the child. For example, babies born vaginally come into contact with the microbiota of the vagina and with maternal feces, and colonization of the neonatal intestine will occur by microbes associated with the vagina, which causes colonization of the neonatal intestine by *Lactobacillus* and *Prevotella* [190]. On the contrary, newborns born by caesarean section are directly exposed to environmental microorganisms by the maternal skin, hospital staff, or hospital environment. In the latter children, a poor and incomplete microbiota has been described, with reduction of the taxa *Bifidobacterium* and *Bacteroides* and an increase in *Clostridium difficile* [191]. The differences in microbiota observed in babies delivered vaginally and those born by caesarean section are evident in their long-term state of health. In fact, the second category has a reduced level of various cytokines and an increased risk of immune disorders [192]. Another fundamental factor responsible for early intestinal colonization is the infant feeding mode. Breastfed children possess a mix of nutrients and promicrobial and antimicrobial agents, which favors the development of a so-called “milk-oriented microbiota” compared to those fed with infant formula. In addition, breast milk contains IgAs that promote a more “tollerogenic” immune system, as well as milk oligosaccharides that can influence the growth of beneficial microbes [193,194]. Artificially breastfed babies are exposed to different carbohydrates, bacteria, and nutrients, generating different patterns of microbial colonization in the intestine [195]. Several studies were conducted to assess whether the mode of birth (natural or caesarean) is involved in the occurrence of MS, but the results were contradictory. Conversely, Dalla Costa et al. have demonstrated that the type of delivery and the type of lactation affect the age of onset of MS. In particular, caesarean section and formula milk anticipate the age of onset of the disease [196]. Interpretation of these results could be influenced by the fact that both caesarean section and formula milk lead to an imbalance in the development of the microbiota, which would participate in the alteration of the immune system, culminating in the onset of an autoimmune pathology such as MS. If the cause–effect sequence between microbiota alteration and MS could be identified, this pathology might be considered as the result of altered communication that starts in the intestine and reaches the brain, and new pharmacological strategies could also be devised [197]. The need to block the advancement of MS and its associated disability is becoming a priority. For this reason, we suggest early treatment of the disease with the current drugs, which are much more effective than the pharmacological protocol used up to two decades ago [198]. In particular, drug treatment should be started immediately after diagnosis, even in the absence of obvious progress of the disease, so as to minimize long-term harm to people with MS [199]. The best strategy could lengthen remission, reduce the onset of progressive forms, slow the progression of the disease, and retain residual function [200]. With this expectation, it would also be interesting to reduce the effects of additional risk factors involved in MS. Since intestinal dysbiosis has been strongly correlated with this neurological disease [201,202], it would be beneficial to adopt additional strategies, along with early drug treatment, so as to further slow progression. To this end, acemannans, contained in *Aloe vera* gel and responsible for a positive change in the composition of the gut microbiota, could be considered good prebiotic adjuvants in the treatment of MS. In addition, it would be advisable to also test the fibers contained in bergamot fruit, to evaluate and confirm the same function. In light of the above, further studies are needed to support this hypothesis, with the expectation of improving the quality of life of people affected by MS and changing the historical view of a two-stage disease.

## Figures and Tables

**Figure 1 nutrients-14-02711-f001:**
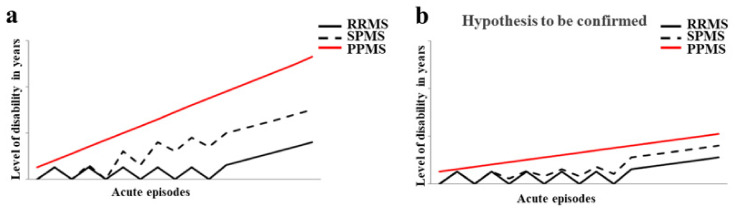
The trend of disability in the different forms of MS. In panel (**a**) the evolution of disability in RRMS, SPMS, and PPMS is shown. Panel (**b**) highlights how the disease could evolve if treated early with the currently validated drugs.

**Figure 2 nutrients-14-02711-f002:**
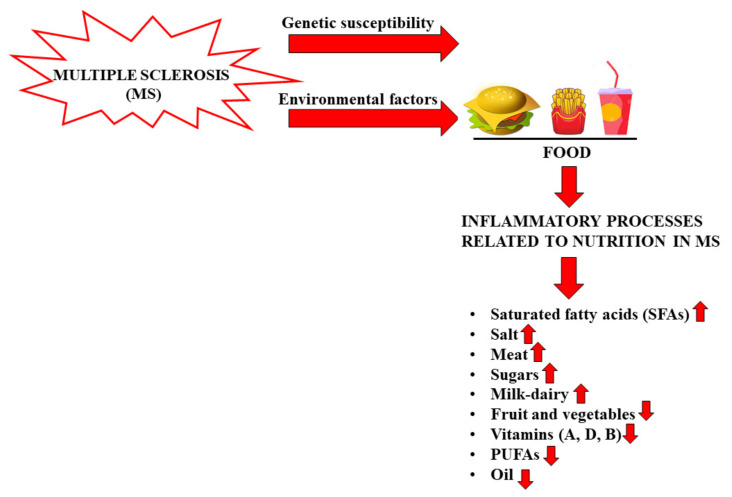
Food can increase or reduce inflammatory processes in MS.

**Figure 3 nutrients-14-02711-f003:**
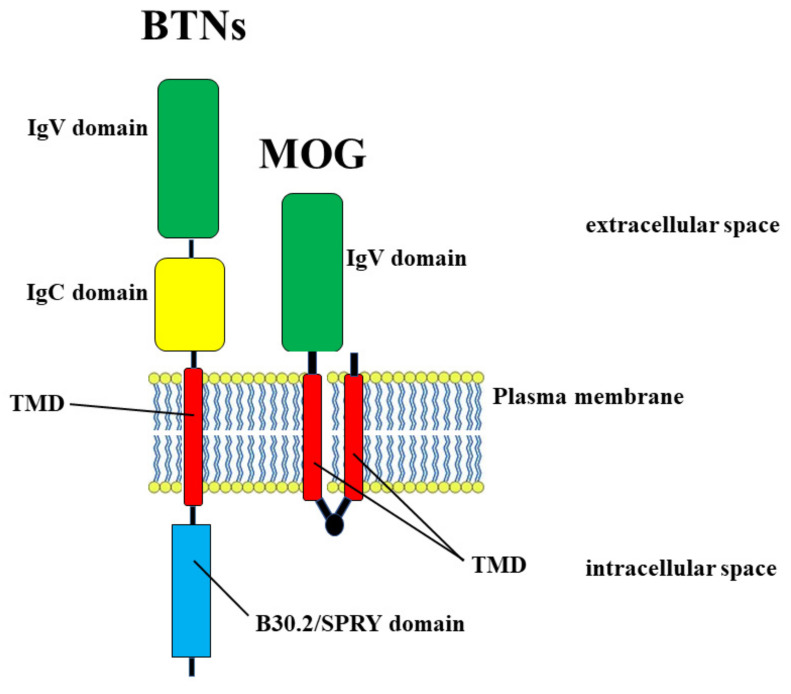
Molecular mimicry between BTN and MOG proteins that have in common the domains IgV and TMD.

**Figure 4 nutrients-14-02711-f004:**
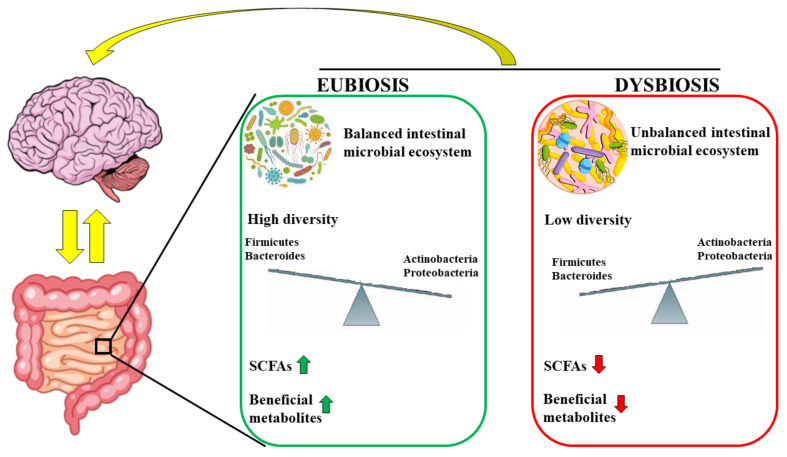
The microbiota is in constant correlation with the brain in conditions of eubiosis and dysbiosis. In particular, the eubiosis is characterized by balanced intestinal microbial ecosystem, high concentration of optimal bacteria (*Actinobacteria* and *Proteobacteria*), high levels of SCFAs and beneficial metabolites. On the contrary, dysbiosis is represented by unbalanced intestinal microbial ecosystem, high concentration of negative bacteria (including *Firmicutes* and *Bacteroides*), low levels of SCFAs and beneficial metabolites.

**Figure 5 nutrients-14-02711-f005:**
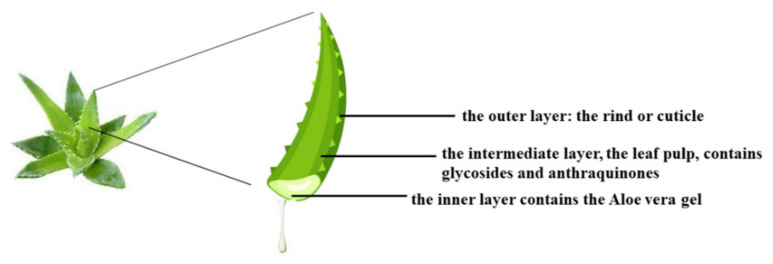
The main sections of the *Aloe vera* leaf.

**Figure 6 nutrients-14-02711-f006:**
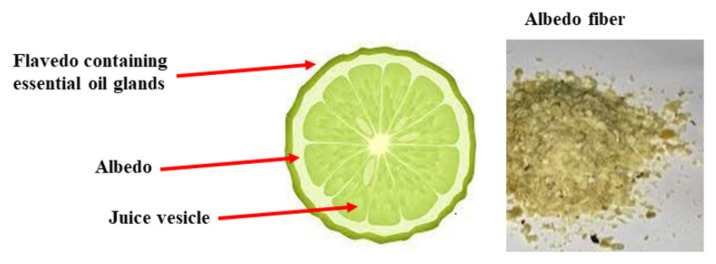
Description of bergamot and image of the fiber obtained from albedo.

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
