# Peer review of "Involvement of the Intestinal Microbiota in the Appearance of Multiple Sclerosis: Aloe vera and Citrus bergamia as Potential Candidates for Intestinal Health"

_nutrients, 2022, doi:10.3390/nu14132711_

Round 1

Reviewer 1 Report

There are some problems in the topic of this manuscript entitled “Involvement of the gut microbiota in the onset of multiple 2 sclerosis: prebiotic role of Aloe vera and Citrus bergamia”. Up to now, there is little causal evidence that Aloe Vera and citrus bergmia improve multiple 2 sclerosis via gut microbiota. Therefore, it is difficult for the author to provide high-quality arguments and conclusions on the subject. Several conclusions in the abstract are common sense.

For example:

1,The gut of mammals is hosting a microbial community of about 2,000 20 bacterial species called the "microbiota", whose composition changes over the life of each individual.

2,In healthy adults, the microbiota is primarily composed of five bacterial phyla: Firmicutes 22 (79.4%), Bacteroidetes (16.9%), Actinobacteria (2.5%), Proteobacteria (1%) and Verrucomicrobia (0.1%).

3, When gut microbiota consists of a high diversity and abundance of microbial populations, this con-24 dition is known as "eubiosis". On the contrary, the state of "dybiosis" refer to altered diversity and 25 abundance of the microbiota.

Although the author tried to describe the relationship between the type of intestinal microflora and the progression of Multiple sclerosis, most evidences the author cited are based on speculation, for example: In particular, natural extracts from Aloe Vera and Bergamot fruits, rich in poli-30 phenols and with an high percentage of polysaccharides, (mostly found in indigestible and fermentable fibers), appear to be potential candidates to re-equilibrate gut microbiota in MS patients. The present review article aims to assess the pathophysiological mechanisms that highlight the role of   microbiota in the development of MS. In addition, the potential for supplementing patients undergoing early stages of MS with Aloe Vera as well as Bergamot fibers on top to the use of conventional 35 drug treatments is discussed.

And the scientific quality of the graph is poor.

Author Response

Dear reviewer,

thank you for your valuable tips and advice.

In the revised manuscript, I tried to highlight the role of Aloe vera and Citrus bergamia as hypothetical and potential and I left open the possibility that appropriate clinical trials can be carried out in order to demonstrate the efficiency of these candidates.

I previously discussed the known importance in the literature of acemannans and fibers on the microbiota (lines 455-464). So being my candidates rich in these components, I propose that they can be used. In fact, the title has also been changed to “Involvement of the intestinal microbiota in the appearance of multiple sclerosis: Aloe vera and Citrus bergamia as potential candidates for intestinal health”.

In this direction, I updated abstracts, the purpose of the paper in the introduction and discussion.

I tried to improve the quality of the figures.

Regards,

Jessica Maiuolo

Reviewer 2 Report

This is a clearly written review that can be relevant for the audience to read. I would be accepting the manuscript after minor revision:

·       The first part of the manuscript which reviews the effects of nutrition and microbiome on multiple sclerosis is well-organized and provides enough information to be considered of interest. My main concern comes from the second part about Aloe vera and Citrus bergamia. The authors do not include any scientific evidence of an effect of these two prebiotics on multiple sclerosis patients or animal models. As in a review, they should include results from previous scientific publications on this regard, instead of just speculate about the effects of Aloe vera and Citrus bergamia as prebiotics to improve multiple sclerosis. In case no studies have been published supporting this idea, the authors should clarify that, and then indicate why, in particular, do they choose to propose these two potential prebiotics instead of other for this review.

·       Aside from this, some typo mistakes were found throughout the review such as:

o   Line 156: “caractrerized” 

o   Line 442: “microbipta”

o   Lines 478-488 and 655: the font size seems smaller.

o   Lines 385, 469, 523, 661: Double space between words.

Author Response

(The authors gave the same response as above.)

Round 2

Reviewer 1 Report

1. Please add authoritatively causal evidences.

2. Please simplify references and contents.

3. Please add mechanism information to figures and optimize color matching to improve figures quality.

Author Response

Dear reviewer,

the following changes have been made to the manuscript: references and contents have been simplified (224 to 203 references) but, at the same time, causal evidences have been added.

The figures have been updated and their captions have been enriched.